# The Relationship between Renin–Angiotensin–Aldosterone System (RAAS) Activity, Osteoporosis and Estrogen Deficiency in Type 2 Diabetes

**DOI:** 10.3390/ijms241511963

**Published:** 2023-07-26

**Authors:** Bongeka Cassandra Mkhize, Palesa Mosili, Phikelelani Sethu Ngubane, Ntethelelo Hopewell Sibiya, Andile Khathi

**Affiliations:** 1Human Physiology, Health Science, Westville Campus, University of KwaZulu-Natal, Westville 4041, South Africa; 215032519@stu.ukzn.ac.za (B.C.M.); 215065077@stu.ukzn.ac.za (P.M.); ngubanep1@ukzn.ac.za (P.S.N.); 2Pharmacology Division, University of Rhodes, Grahamstown 6139, South Africa; n.sibiya@ru.ac.za

**Keywords:** renin–angiotensin–aldosterone system, osteoporosis, estrogen deficiency, type 2 diabetes

## Abstract

Type 2 diabetes (T2D) is associated with a plethora of comorbidities, including osteoporosis, which occurs due to an imbalance between bone resorption and formation. Numerous mechanisms have been explored to understand this association, including the renin–angiotensin–aldosterone system (RAAS). An upregulated RAAS has been positively correlated with T2D and estrogen deficiency in comorbidities such as osteoporosis in humans and experimental studies. Therefore, research has focused on these associations in order to find ways to improve glucose handling, osteoporosis and the downstream effects of estrogen deficiency. Upregulation of RAAS may alter the bone microenvironment by altering the bone marrow inflammatory status by shifting the osteoprotegerin (OPG)/nuclear factor kappa-Β ligand (RANKL) ratio. The angiotensin-converting-enzyme/angiotensin II/Angiotensin II type 1 receptor (ACE/Ang II/AT1R) has been evidenced to promote osteoclastogenesis and decrease osteoblast formation and differentiation. ACE/Ang II/AT1R inhibits the wingless-related integration site (Wnt)/β-catenin pathway, which is integral in bone formation. While a lot of literature exists on the effects of RAAS and osteoporosis on T2D, the work is yet to be consolidated. Therefore, this review looks at RAAS activity in relation to osteoporosis and T2D. This review also highlights the relationship between RAAS activity, osteoporosis and estrogen deficiency in T2D.

## 1. Introduction

The renin–angiotensin–aldosterone system (RAAS) is a fluid and electrolyte regulatory system responsible for the maintenance of blood volume and pressure [1]. This system has been studied extensively and has been evidenced to be crucial in maintaining glucose homeostasis [2]. A plethora of studies have highlighted the relationship between hyperglycemia and alterations in RAAS, and the development of estrogen deficiency and osteoporosis [3,4]. RAAS components have been identified in bones, whereby the interaction between angiotensin II (Ang II) and the angiotensin II type 1 receptor (AT1R) inhibits osteoblast maturation [5]. Furthermore, upregulation of Ang II has been reported to elevate nuclear factor-κB ligand (RANKL) and decrease osteoprotegerin (OPG) expression, thus activating osteoclasts, which promote bone resorption [6]. Additionally, the upregulation of Ang II promotes the upregulation of aldosterone, which is another RAAS hormone that affects bone turnover by binding to mineralocorticoid receptors (MR) in osteoclasts, osteocytes and osteoblasts [7,8]. The Ang II/AT1R axis has been reported to stimulate proinflammatory cytokines that promote bone resorption and hinder osteoblast differentiation [9]. MRs are also expressed in the parathyroid tissue and several authors have demonstrated a positive correlation between the concentrations of aldosterone and serum parathyroid hormone (PTH), a calcium-regulating hormone [10]. Collectively, upregulation of the Ang II/AT1R interaction, the shift in the OPG/RANKL ratio and the increase in aldosterone contribute to decreased bone mineral density (BMD) [11], consequently altering the microarchitecture of the bone structure and leading to the development and progression of osteoporosis [11]. Other RAAS components such as angiotensin 1-7 (Ang 1-7), angiotensin 1-9 (Ang 1-9) and angiotensin-converting enzyme 2 (ACE2) are expressed in bone tissue and have been reported to negate the effects of Ang II/AT1R interaction and RANKL upregulation [12,13]. Furthermore, estrogen is a well-known bone anti-resorption hormone [14]. Estrogen exerts numerous physiological effects that maintain glucose homeostasis [15]. However, estrogen deficiency is associated with impaired glucose uptake, impaired insulin secretion, insulin resistance, increased gluconeogenesis and increased lipolysis, which are clinical markers of T2D [16,17]. Interestingly, RAAS is reportedly upregulated in T2D; therefore, this review seeks to highlight the relationship between RAAS and osteoporosis and their association with estrogen deficiency in T2D.

### 1.1. The Renin–Angiotensin–Aldosterone System

The renin–angiotensin–aldosterone system (RAAS) is a regulatory signaling pathway that maintains fluid and electrolyte homeostasis [18]. Renin is secreted by the kidneys in response to a decrease in blood pressure, blood volume, plasma sodium and potassium levels [19]. Renin has also been shown to induce the release of prorenin from the juxtaglomerular cells in the kidneys [20]. The release of prorenin results in a cascade of events where inactive angiotensin I (Ang I) is produced from angiotensinogen (AGT), which is subsequently cleaved by angiotensin-converting enzyme (ACE) into angiotensin II (Ang II). Ang II activates Ang II type 1 (AT1), which is responsible for inducing vasoconstriction in the systemic vasculature, subsequently raising blood pressure [21]. Furthermore, Ang II acts on the adrenal cortex to synthesize aldosterone, which promotes sodium and water reabsorption through activation of the mineralocorticoid receptor in the nephrons [2,21]. The renin–angiotensin–aldosterone system is pathologically activated in type II diabetes; this has been noted to result in various detrimental effects, including osteoporosis and estrogen deficiency [22,23,24]. Recent studies have evidenced the presence and activity of the renin–angiotensin system in T2D in various tissues, with observable changes in angiotensinogen, renin, ACE, aldosterone, angiotensin II, AT1R, AT2R, Ang 1-7 and Ang 1-9 in bones [22,25]. Furthermore, patients who suffer from T2D and hypertension usually have osteoporosis, with this phenomenon being highly prevalent in postmenopausal women [26].

### 1.2. Local RAAS and T2D

Several studies have highlighted the role of RAAS in the development and progression of insulin resistance, as RAAS is produced locally in numerous organs [27]. RAAS is upregulated in the skeletal muscle, which is a major site of glucose utilization [28]. Ang II suppresses phosphorylation of insulin receptor substrate (IRS)-1 in muscles, blocking increases in phosphatidylinositol 3 (PI3)-kinase and the consequent translocation of glucose transporter (GLUT) 4 to the cell membrane [29]. This results in hyperglycinemia due to impaired glucose uptake and insulin resistance [30]. Hyperglycemia induces p53 glycosylation, which has been implicated in angiotensinogen transcription and the subsequent generation of Ang II from the local RAAS [31], thus promoting the upregulation of local RAAS in various organs [25]. Furthermore, the upregulation of RAAS induces insulin resistance in adipose tissue [30]. In T2D, the renin-encoding gene is expressed and upregulated in adipose tissue, which also contains Cathepsin D and G enzymes that can create Ang I and Ang II via RAAS alternate pathways [32]. Nutrition was proposed as the link between RAAS upregulation and hyperglycemia. An animal study evidenced the upregulation of RASS mRNA, translocation of the RAAS genes and expression of RAAS proteins in rats fed on high-fat diets and high-fat-high-carbohydrate diets [33]. Angiotensin II induces the formation of prostacyclin in adipose tissue, which causes the conversion of preadipocytes to adipocytes and increases lipid synthesis and storage in adipocytes through Ang II/AT2R signaling and the ACE2/Ang 1-7/Mas receptor [34]. However, in overt hyperglycemia, which occurs in T2D, the expression of Ang II and AT1R is upregulated whilst the expression of AT2R and ACE2/Ang 1-7/Mas receptor is downregulated [28]. Ang II/AT1R signaling has been reported to induce lipogenesis and impair the differentiation of preadipocytes, thus affecting adipose storage capacity [28,33]. Consequently, the adipose tissue is overloaded with lipids, resulting in the ectopic redistribution of fats to other organs [33]. Additionally, a positive correlation has been noted between insulin resistance and the upregulation of Ang II/AT1R signaling [35]. Hence, the upregulation of adipose RAAS contributes to the development and progression of insulin resistance and T2D [35]. The increased lipogenesis contributes to the storage of fats in the liver [36]. Hypertriglyceridemia is closely associated with insulin resistance [37]. The liver and adipose tissue are the main sites of triglyceride synthesis [38]. Ang II/AT1R signaling, which is upregulated in T2D, is involved in the suppression of the protein expression and enzymatic activity of hepatic N-deacetylase (NDST) [39] This is a key regulatory enzyme involved in heparan sulphate (HS) biosynthesis in the diabetic state and suppression of hepatic NDST was suggested to lead to diabetic dyslipidemia [40,41,42]. Furthermore, Ang II-based activation of AT1R causes insulin resistance by increasing hepatic triglyceride levels, which is thought to contribute to the development of diabetes [28,43]. Diabetic dyslipidemia is associated with damage to vital organs such as the heart and kidneys [44,45]. Local RAAS in the kidneys has been evidenced, in various studies, to cause morphological changes that affect kidney function [44,46]. In T2D, systemic and kidney RAAS have been positively correlated with proteinuria and hypertension [46,47]. Increased blood pressure has been evidenced to cause stress on the heart, thus affecting its function [29,48]. Interestingly, local RAAS activity has also been reported to compromise the integrity of the heart [28,49]. In T2D, activation of the Ang II/AT1R pathway can promote cell growth and proliferation, apoptosis, oxidative stress generation, inflammation and fibrosis, which can contribute to cardiac remodeling and atherosclerosis [50,51]. Therefore, research has been focused on this system due to the detrimental effects of the upregulation of ACE/Ang II/AT1R signaling and downregulation of the AT2R and Ang 1-7/AT2R/Mas axis in the various organs (Figure 1) [28]. A study by Zheng and colleagues showed an improvement in insulin resistance and the mentioned downstream changes when the ACE/Ang II/AT1R axis was blocked [35]. Recently, local RAAS has been reported in bones and associated with bone fragility and the development of osteoporosis in T2D due to the shift in pro-resorptive and anti-resorptive factors in favor of pro-resorptive factors [52,53].

## 2. Pro-Resorptive Factors

### 2.1. Parathyroid Hormone

Parathyroid hormone (PTH) is an 84-amino acid polypeptide released by the parathyroid glands in response to calcium deficiency [54]. PTH promotes renal tubular calcium reabsorption and stimulates renal 1,25 dihydroxy vitamin D synthesis, thus indirectly enhancing intestinal calcium absorption and bone remodeling [54]. PTH is critical in calcium homeostasis and the consequent formation of bone integrity by increasing the number of bone-forming cells as well as stimulating osteoblast development and reducing osteoblast cell death [54,55]. The parathyroid hormone further contributes to calcium homeostasis by promoting bone resorption when there is a calcium deficiency [55]. Studies have evidenced the expression of AT1R and mineralocorticoid receptors (MR) in the parathyroid glands [56,57]. Studies have shown that RAAS, along with the expression of AT1R and MR receptors, is upregulated during T2D, thus enhancing PTH release from the parathyroid glands [56,57]. Due to chronic hyperglycemia in T2D, the parathyroid glands are overstimulated, resulting in excessive release of PTH [58]. The increased PTH levels result in excessive bone resorption, which affects the structural integrity of the bone [58]. Hence, the upregulation of RAAS in T2D may stimulate the production of excessive amounts of PTH, leading to bone fracture and osteoporosis [3]. However, more studies are required to generate evidence for this theory. PTH is pivotal for calcium homeostasis as it promotes the production of 1,25-hydroxy vitamin D in the kidneys [59]. 1,25-hydroxy vitamin D restores calcium balance by promoting bone resorption, and the restored calcium homeostasis inhibits the release of PTH [59]. MR and AT1R are expressed in the parathyroid glands [56]. T2D leads to upregulation of aldosterone and the ACE/Ang II/AT1R axis, potentially resulting in the parathyroid glands being over-stimulated and leading to increased PTH and 1,25-hydroxy vitamin D concentrations that could further lead to osteoporosis [3,50,59,60]. Furthermore, PTH stimulates the release of aldosterone from the adrenal glands due to the expression of type 1 PTH receptors in the adrenal glands, indicating a cycle of continued activity [56].

### 2.2. Vitamin D3

Vitamin D3 plays a key role in calcium homeostasis and can be synthesized endogenously in the skin or absorbed via the intestines from our diet, converted to 25-hydroxyvitamin D (25(OH)_2_D) in the liver or to the bioactive form, 1,25-hydroxyvitamin D (1,25(OH)_2_D), in the kidneys [61]. In the postnatal stage, one of the principal functions of 1,25(OH)_2_D is to maintain calcium homeostasis by enhancing calcium absorption in the intestines [62]. When dietary calcium levels are low, 1,25(OH)_2_D increases transcellular intestinal calcium transport by upregulating the expression of the apical membrane calcium channel transient receptor potential vanilloid 6 (TRPV6) and the calcium-binding protein calbindin-D9k as these channels promote calcium reabsorption [61,63]. Additionally, 1,25(OH)_2_D promotes distal tubular calcium reabsorption in the kidney [64]. T2D causes renal morphological and functional alterations due to the upregulation of systemic and renal RAAS via the ACE/Ang II/AT1R axis, consequently affecting calcium homeostasis [58]. Calcium loss promotes an increase in 1,25(OH)_2_D, inducing bone breakdown—a process known as bone resorption—in an attempt to restore blood calcium levels [65]. A continued state of bone resorption contributes to bone fragility, increased risk of bone fracture and osteoporosis [66]. Interestingly, a recent study suggested an inverse relationship between 1,25(OH)_2_D and plasma renin, whereby 1,25(OH)_2_D has an inhibitory effect on renin [10]. However, PTH promotes aldosterone synthesis in the adrenal glands. Studies show that changes in PTH and 1,25(OH)_2_D are largely influenced by systemic and renal RAAS, affecting kidney function; however, growing evidence shows an interaction between local RAAS, inflammatory cytokines and bone remodeling [67,68]. Recent studies have demonstrated a positive correlation between PTH and 1,25(OH)_2_D and the production of Ang II and aldosterone [67,68]. Furthermore, MRs have not only been identified on the parathyroid glands but they have also been identified on osteoblasts, osteoclasts and bone cells, indicating that aldosterone has a direct effect on bone metabolism (Figure 2) [3]. Physiologically, 1,25(OH)_2_D interacts with the Wnt signaling cascade, which regulates osteoblast differentiation, to induce bone formation and mineralization in osteoblasts as well as osteogenic differentiation from human mesenchymal stem cells [69]. The promoter region of the gene encoding LRP5 is upregulated by interaction with 1,25(OH)_2_D–VDR, increasing LRP5 expression [70]. Interestingly, RAAS, through the ACE/Ang II/AT1R axis, has been evidenced to inhibit the Wnt signaling cascade [71]. RAAS affects 1,25(OH)_2_D, thus leading to increased osteoclast formation [12,72]. Furthermore, the 1,25(OH)2D–VDR interaction is impaired in T2D, further reducing bone formation and mineralization [73]. PTH, 1,25(OH)_2_D, Ang II, glucocorticoids and proinflammatory cytokines have been evidenced to cause osteoclastogenesis, leading to bone remodeling through the expression of nuclear factor-κB ligand (RANKL) [74,75,76,77]. The differentiation and bone-resorbing abilities of the osteoclast depend on RANKL and its receptor, nuclear factor-B (RANK) [77]. Osteoprotegerin (OPG) binds to RANK and inhibits RANKL–RANK binding via competitive inhibition, hence the OPG/RANKL ratio is a measure of osteoclast differentiation [77,78,79]. RANKL influences the immune system and regulates bone remodeling and regrowth [79].

### 2.3. Proinflammatory Cytokines

Osteoblasts and osteoclasts, which are bone-forming and bone-resorbing cells, respectively, are necessary for maintaining bone homeostasis [80]. Osteoblasts synthesize proteins for the bone matrix and encourage bone deposition and mineralization [81]. Bone-related disorders such as postmenopausal osteoporosis, hyperparathyroidism and osteopetrosis are caused by an imbalance between osteoclast and osteoblast activity during bone remodeling as a result of proinflammatory cytokines, growth factors and hormones [82]. Proinflammatory cytokines such as interleukin-1β (IL-1β), interleukin-6 (IL-6) and tumor necrosis factor-α (TNF-α) are involved in the breakdown of bone through enhancement of the development of osteoclast precursors into adult osteoclasts [83]. Furthermore, TNF-α promotes osteoclastogenesis by increasing RANKL expression in osteoblasts [83]. RANKL binding to RANK commits monocytic precursor cells to the osteoclastic lineage, thus promoting bone loss [83]. TNF-α is also involved in osteoclastogenesis through modulation of the wingless-related integration site (Wnt) signaling pathway, which is considered a key regulatory pathway for bone formation by osteoblasts [84,85]. TNF-α is a potent inducer of the production of Dickkopf-1 (Dkk-1), a Wnt antagonist that hinders the development of bone cells (Figure 3) [85,86]. Systemic and local RAAS is upregulated in T2D; hence, the ACE/Ang II/AT1R axis stimulates the activity of proinflammatory cytokines, resulting in increased osteoclastogenesis, reduced bone density and development of osteoporosis [87].

### 2.4. ACE/Ang II/AT1R Axis and ACE2/Ang 1-7/Mas Receptor

Various components of RAAS are expressed in various bone cells and chondrocytes of epiphyseal plates under physiological environments, implicating local RAAS in epiphyseal elongation during bone development and repair [3,88]. Furthermore, a locally active RAAS influencing hematopoietic cell growth, generation, proliferation and differentiation has been identified in bone marrow cells (BMCs), hematopoietic-lineage BMCs and cultured marrow stromal cells (MSCs) [3,13,88]. Ang II/AT1R signaling in these cells affects the production of red blood cells and blood flow in capillaries in the bone marrow [89]. Ang II diminished osteoblastic differentiation and mineralization and reduced the percentage of mineralized nodules by activating AT1R in osteoblast UMR-106 cells [90]. The ACE/Ang II/AT1R axis stimulates bone marrow mononuclear cells to differentiate into multinuclear cells [91,92]. This axis also stimulates multinuclear cells to differentiate into osteoclasts and enhances tartrate-resistant acid phosphatase (TRAP)-positive multinuclear osteoclasts [91]. Osteoclasts are cells that mediate bone loss by increasing their resorptive activity and causing bone degradation, leading to the initiation of the normal bone remodeling process [93]. In pathological conditions such as T2D, systemic and local RAAS are upregulated, particularly the ACE/Ang II/AT1R axis, which may promote osteoclast over-production leading to bone degradation [25,28,94]. Chronic upregulation of RAAS, as observed in T2D, may result in continued bone breakdown, a decline in bone density, osteoporosis and an increased risk of bone fracture. Osteoporosis interventions have focused on RAAS blockers [95]. As previously established, osteoporosis occurs due to an imbalance in bone remodeling, which is marked by a decrease in osteoblastic activity and an increase in osteoclastic activity, resulting in a greater rate of bone resorption [66]. Ang II has been evidenced to play a key role in osteoclastogenesis, and thus osteoporosis, through various mechanisms such as those discussed in the subsequent sections [96].

#### 2.4.1. Ang II and Expression of RANKL

A study demonstrated that Ang II resulted in a significant (eightfold) increase in RANKL mRNA expression through the ERK pathway in human osteoblasts and UMR-106 cells [97]. It also promoted the differentiation of mesenchymal stem cells into multinuclear cells, activated mature osteoclasts responsible for bone resorption and increased the number of TRAP-positive multinuclear osteoclasts, and hence, osteoclastogenesis [98,99]. Studies have suggested that an increase in RANKL upon activation of the AT1R–ERK signaling pathway may serve as a mediator of Ang II-induced osteoclast differentiation and activation [96,100]. In order to investigate this theory, Olmesartan (an AT1R blocker) or U0126 (an extracellular signaling kinase pathway MEK/ERK inhibitor) were added to osteoblasts, resulting in the amelioration of the osteoclastogenesis effects of Ang II [101]. Therefore, studies have alluded to ACE/Ang II/AT1R increasing RANKL expression, thus facilitating pathophysiological bone resorption [102].

#### 2.4.2. Ang II Increases Cyclic Adenosine Monophosphate (cAMP)

RANKL and core-binding factor subunit alpha-1 (Cbfa1/Runx2), both of which are controlled by cAMP, are the main regulators of the differentiation of osteoblast and osteoclast cells [103]. Cbfa1/Runx2 is a transcription factor required for osteoblast development from mesenchymal progenitors and subsequent bone matrix mineralization [103,104]. Furthermore, Cbfa-1 is a potent transcription factor that regulates the expression of many of the osteoblast and chondrocyte functions and triggers the expression of major osteoblast-specific lineage genes [104,105]. Angiotensin II can promote an increase in intracellular cAMP and subsequently activate signaling pathways that are involved in the control of low-density-lipoproteins (LDLs), which in turn changes Cbfa1 expression [106,107]. LDL-mediated cholesterol administration improves osteoclast survival, and hence their lifespan, considerably [108,109]. Therefore, Ang II modifies Cbfa1 expression, consequently reducing the number of osteoblasts and resulting in poor bone formation by stimulating the cAMP signaling pathway and regulating other downstream targets such as LDLs [110,111]. Ang II triggers prostaglandins, which stimulate cAMP signaling, leading to upregulated expression of LDL receptors [112]. The upregulated expression of LDL receptors in the bone marrow leads to increased LDLs, which have been reported to take up space in the marrow, thus affecting osteoblast availability [113].

Furthermore, in vitro studies showed that free cholesterol reduced the proliferation and differentiation of osteoblasts and inhibited the expression of BMP2 and core binding factor alpha 1 (Cbfa1). As mentioned, local RAAS exists in other organs such as adipose tissue, where it alters adipose buffering capacity, thus increasing free cholesterol levels. Free cholesterol increases the level of malondialdehyde (MDA) and decreased the activity of superoxidase dismutase (SOD) in osteoblasts, indicating oxidative stress in the bone microenvironment. Oxidative stress inhibits and decreases osteoblast development and activity, which in turn reduces mineralization and osteogenesis. Furthermore, the increase in cAMP as a result of Ang II activates downstream signaling pathways that, in turn, downregulate the expression of Cbfa1/Runx2 while increasing the expression of RANKL [114]. The decrease in Cbfa1/Runx2 and the increase in RANKL decrease the quantity and function of osteoblasts, resulting in increased bone resorption and decreased bone formation [114].

#### 2.4.3. Ang II Upregulates SOST Gene Expression

Ang II increases the expression of the SOST gene in osteocytes via activation of AT1R [5,85,90]. Sclerostin coding gene (SOST) encodes the secretory protein sclerostin that binds to LRP5/6 (low-density lipoprotein receptor-related protein) receptors on osteoblast cell membranes, inhibiting Wnt/b-catenin signaling and lowering osteoblastic bone production [85,115].

As already mentioned, Ang II/AT1R signaling is upregulated in T2D, resulting in the above cascade of events [90]. Therefore, local bone RAAS, specifically Ang II, has been positively associated with decreased bone density, increased bone turnover and impaired bone microarchitecture, leading to the development and progression of osteoporosis [116]. However, the ACE2/Ang 1-7/MasR receptor arm of bone RAAS has been demonstrated to increase the expression of OPG, a glycoprotein that regulates bone remodeling [13]. OPG regulates bone remodeling by controlling osteoclast activity, hence interfering with the interaction between RANK and its ligand RANKL [76]. In order to regulate cell proliferation, the apoptosis regulator gene RANKL interacts with OPG, a ligand for the RANK receptor, under physiological conditions (Figure 4) [76]. However, the ACE/Ang II/AT1R axis is upregulated and the ACE2/Ang 1-7/MasR arm is downregulated in T2D, consequently decreasing the expression of OPG protein [13,117]. Thus, the shift in the RANKL/OPG ratio is regarded as one of the markers of bone fragility and the development of bone disorders such as osteoporosis [13].

### 2.5. ANG 1-7

The ACE2/Ang1-7/Mas receptor cascade is thought to be the beneficial arm in the biological consequences of systemic and local RAAS. Ang (1-7) improves abnormal bone metabolism and micro-architecture [12,13,118]. Furthermore, Ang (1-7) significantly increases mineralization while inhibiting osteoclastogenesis [12,118]. To test this theory, A-779 was used to block the Mas receptor, resulting in significantly reduced beneficial effects of Ang (1-7) on bone health, indicating that the Mas receptor plays a critical role in regulating Ang (1-7)’s osteoprotective properties [13,119,120]. The role of ACE/Ang II/AT1R in osteoclastogenesis due to the activated proinflammatory cytokines has been discussed [96]. Interestingly, evidence suggests that the ACE2/Ang1-7/Mas receptor axis counteracts the proinflammatory cytokine effects, thus improving bone health. Ang-(1-7) reduces the expression of proinflammatory cytokines associated with bone resorption, namely IL-6 and TNF-α [121,122,123,124]. In osteoporosis-related alveolar bone resorption, IL-6—an osteoclastogenesis-promoting cytokine—promotes the conversion of forkhead box P3 (FoxP3) T cells to T helper 17 (Th17) cells, which guard against germs but induce osteoclast formation and thus, bone injury [125,126,127]. Additionally, the ACE2/Ang1-7/MasR axis has been observed to reduce the expression of RANKL and IL-1β mRNA levels, hence reducing osteoclastogenesis (Figure 5) [12,124]. However, in T2D, the conventional RAAS axis is upregulated and the ACE2/Ang1-7/MasR axis is downregulated and may thus compromise bone health [124,128,129]. Anti-resorptive factors such as calcium, estrogen and TGFb promote bone health via the ACE2/Ang1-7/MasR axis and the Ang (1-7)/AT2R pathway [13,130].

## 3. Anti-Resorptive Factors

### 3.1. Transforming Growth Factor β (TGF-β)

Osteoblasts develop from mesenchymal stem cells (MSCs) in the bone marrow, while osteoclasts develop from hematopoietic stem cells (HSCs) [131,132]. TGF-β1 regulates both osteoblast and osteoclast differentiation, thus balancing bone production and resorption [133]. TGF-β1 maintains bone density by promoting osteoblast proliferation, inhibiting osteoblast apoptosis and attracting osteoblastic precursors or matrix-producing osteoblasts to the location via chemotactic attraction [133,134]. Additionally, in the first phases of osteoblast development, TGF-1β increases the synthesis of extracellular bone matrix protein by osteoblasts, thus maintaining bone health [134]. Estrogen increases osteoblast proliferation and differentiation by increasing the synthesis of TGF- β1 in osteoblasts [135]. In addition, estrogen inhibits bone loss by encouraging osteoclast apoptosis through a TGF-dependent process [136]. However, in T2D, there is an estrogen deficiency associated with impaired estrogen receptor expression that results in reduced TGF-β1 [14,137,138]. Hence, estrogen deficiency due to the upregulation of RAAS is positively correlated with decreased TGF-β1 expression, leading to bone loss [139,140]. Furthermore, studies have highlighted a positive correlation between increased glucocorticoids and T2D [141,142]. Glucocorticoids are known to promote the apoptosis of osteoblasts and inhibit their proliferation and differentiation while promoting the differentiation of osteoclasts [143,144]. Glucocorticoids upregulate TGF-β1 expression in osteoblasts; however, unlike estrogen, glucocorticoids synergize with TGF-β1 to enhance osteoclast formation by stimulating the priming of osteoclast progenitors for differentiation into osteoclasts [145]. Therefore, the upregulated glucocorticoids contribute to the progression of insulin resistance to the development of T2D due to increasing glucose levels [141,146]. Furthermore, local bone RAAS activity has been reported in glucocorticoid-induced osteoporosis, hence therapies that target the RAAS system have been reported to improve glucose homeostasis and hence the declining bone health [5,52,76,147]. Numerous cell types, including osteoblasts, B lymphocytes and osteogenic stromal stem cells produce OPG in response to anti-resorptive agents such as estrogen and TGFβ-related bone morphogenic proteins [148,149].

### 3.2. Estrogen

#### 3.2.1. Estrogen Physiology

Estrogen is a sex hormone—predominantly synthesized in the ovaries in females and testes in males—that governs the growth, development and physiology of the reproductive system [150]. The neuroendocrine, skeletal, adipogenic and cardiovascular systems are similarly impacted by estrogen due to the presence of estrogen receptors in these organs [151]. Estrogen is associated with glucose handling as it has been shown to stimulate insulin sensitization in the skeletal muscle and adipose tissue through the insulin signaling pathway [152]. Therefore, estrogen, due to the ERs in the skeletal muscle and adipose tissue, is associated with impaired insulin signaling and the development and progression of hyperglycemia reported in T2D [152,153,154,155].

Studies have determined the presence of estrogen receptor alpha (ERα) and estrogen receptor beta (ERβ) in the bone marrow [156]. Osteoblasts, osteoclasts and osteocytes express ERs, which have beneficial effects on bone integrity [156]. The binding of estrogen to ERs modulates the expression of genes that encode proteins that are estrogen targets, including IL-1β, insulin-like growth factor 1 (IGF1) and TGF-β1 [157,158,159]. As discussed, TGF-β1 promotes osteoblast proliferation, inhibits osteoblast apoptosis and recruits osteoblastic precursors or matrix-producing osteoblasts to maintain bone density [133,149,160]. Furthermore, the binding of estrogen to ER promotes the upregulation of OPG and the downregulation of RANKL, thus impairing osteoclastogenesis and bone resorption [161,162]. Additionally, the estrogen–ER interaction activates Wnt/β-catenin signaling to increase osteogenesis by curbing the differentiation of MSC to adipocytes and facilitating the differentiation of mesenchymal stem cells to osteoblasts, thus promoting bone formation [163]. Sclerostin is a Wnt antagonist that competitively binds to LRP5/6 to inhibit the Wnt signaling pathway [164]. Reports show that sclerostin is significantly increased in postmenopausal women in comparison to premenopausal women [165]. Therefore, due to reduced estrogen levels in postmenopausal women, sclerostin is significantly increased, leading to morphological changes in the bone marrow microarchitecture due to the inhibition of osteoblastogenesis and the increase in bone resorption [165,166,167]. Moreover, reports have suggested that the ACE/Ang II/AT1R axis upregulates the SOST gene that encodes sclerostin, thus promoting osteoclastogenesis [168]. The predominant form of estrogen, estradiol (E2), has been shown to increase the mRNA levels of the AGT gene [24]. Interestingly, E2 has been evidenced to downregulate AT1R expression and reduce renin and ACE activity [169]. Additionally, AT1R mRNA is regulated post-transcriptionally by estrogen-sensitive binding proteins [170,171]. Although estrogen’s effect on Ang II has not been fully elucidated, it can attenuate Ang II responses to AT1R [171]. Estrogen treatment elevates Ang (1-7) levels in transgenic mice [172,173]. Structural and biochemical changes in rats with ovariectomy-induced osteoporosis are improved by the ANG (1-7) axis; therefore, estrogen might use this axis to exert its osteoprotective effects [13,174]. Moreover, estrogen has been evidenced to mediate the transcription of ACE2; thus ACE2 is downregulated and AT1R is upregulated in estrogen deficiency, leading to estrogen deficiency-induced osteoporosis [174]. Ang II also downregulates the mRNA expression of osteocalcin, which is uniquely produced during the maturation of osteoblastic cells [175]. Additionally, Ang II reduces the activity of alkaline phosphatase (ALP), which is a hallmark of osteoblastic differentiation [175,176]. Thus, estrogen is critical for balancing bone resorption and formation [26,177].

#### 3.2.2. Estrogen Deficiency in T2D

A bi-directional relationship between estrogen deficiency and T2D has been studied, where hyperglycemia is linked to reduced estrogen bioavailability and a lack of estrogen is shown to contribute to the progression of insulin resistance [153,178]. Postmenopausal women have been shown to suffer from progressive impaired glucose tolerance, declining bone mass density and increased bone turnover due to estrogen deficiency [179,180]. Furthermore, estrogen regulates the expression of its own receptors, hence estrogen deficiency is associated with reduced ER expression in the bone marrow [181,182]. As previously stated, osteoblasts express ERs. As T2D is associated with estrogen deficiency, ERs are also reduced because estrogen regulates the expression of its own receptors, thus potentially leading to impaired bone metabolism [16,161]. Rats lacking estrogen exhibit an imbalance between the traditional ACE/Ang II/AT1R receptor pathway and the ACE2/Ang1-7/Mas-receptor pathway in their femurs [183,184]. Hence, focused interventions have proven that blocking Ang II action with an AT1R antagonist prevents bone loss associated with estrogen deficiency [185,186]. The hyperglycinemia noted in T2D is positively correlated with estrogen deficiency and the upregulation of the ACE/Ang II/AT1R axis and downregulation of the ACE2/Ang 1-7/Mas receptor axis and AT2R, which may promote osteoporosis [185,186]. Furthermore, due to the effect of estrogen on glucose handling, estrogen deficiency and the upregulation of local RAAS are linked to increased low-density lipoproteins (LDLs) [184,187]. Increased LDLs were shown to correlate with low bone mass in postmenopausal women, alluding to a possible relationship between triglycerides and the maintenance of bone mass (Figure 6) [188,189]. The relationship between estrogen deficiency, increased systemic and local RAAS activity and hypertension has recently received attention [190]. A plethora of studies have evidenced the correlation between systemic RAAS and hypertension [19,190]. Over the past decade, the focus has shifted to local RAAS, including bone RAAS, osteoporosis and hypertension [191]. Data demonstrates that these are enhanced in postmenopausal women who have significantly reduced estrogen levels [24,186]. Hence, a relationship between estrogen deficiency, RAAS activity and osteoporosis has gained traction [186]. Interestingly, hypertension and systemic and local RAAS upregulation has been demonstrated in T2D [51,192]. Hence, the correlation between RAAS, hypertension, osteoporosis and estrogen deficiency in T2D is of interest for formulating effective treatment strategies. Henceforth, this review focuses on the plethora of mechanisms involving RAAS and their association with osteoporosis and estrogen deficiency in association with T2D.

## 4. Conclusions

Hypertension, osteoporosis, adiposity and estrogen deficiency have been identified as risk factors for T2D [193,194,195]. The renin–angiotensin–aldosterone system is positively correlated with T2D and hypertension [196,197,198,199]. Anti-hypertensive medications that block the RAAS have been evidenced to have osteoprotective effects [3,200,201,202,203,204,205]. Furthermore, RAAS blockers are directly correlated with improved glucose handling and inhibition of progressive insulin resistance [199,206]. In this review, the mechanistic action of this system in association with the shift in bone formation and resorption has been highlighted. The effects of the classical arm and the physiologically beneficial arm were analyzed and this system was shown to affect the inflammatory status, subsequently affecting the microarchitecture of the bone [3,207,208]. Additionally, the presence of RAAS receptors, including mineralocorticoid receptors in the parathyroid and adrenal glands and osteoblasts has been evidenced, further emphasizing the role of RAAS in calcium handling and its effects in the bone microenvironment [209,210,211,212].

In addition to the relationship between RAAS, hypertension and osteoporosis, these findings may suggest that the upregulation of RAAS in T2D acts on its receptors and contributes to the development of osteoporosis. This review further focused on the relationship between RAAS, osteoporosis and estrogen deficiency in T2D. Estrogen deficiency has been proposed as a biomarker for T2D due to the critical role of estrogen in glucose handling [16,213]. Additionally, several studies have investigated the role of local and systemic RAAS in relation to hypertension and glucose homeostasis [214,215]. Interestingly, estrogen deficiency has been associated with hypertension and osteoporosis in postmenopausal women. This review focused on the relationship between estrogen deficiency, osteoporosis and hypertension in T2D by exploring mechanistic pathways that may be involved, for example, RAAS.

The association between RAAS and osteoporosis has been identified independently of T2D. However, a study investigating RAAS activity in T2D and its possible role in the development of osteoporosis in T2D is required.

## Figures and Tables

**Figure 1 ijms-24-11963-f001:**
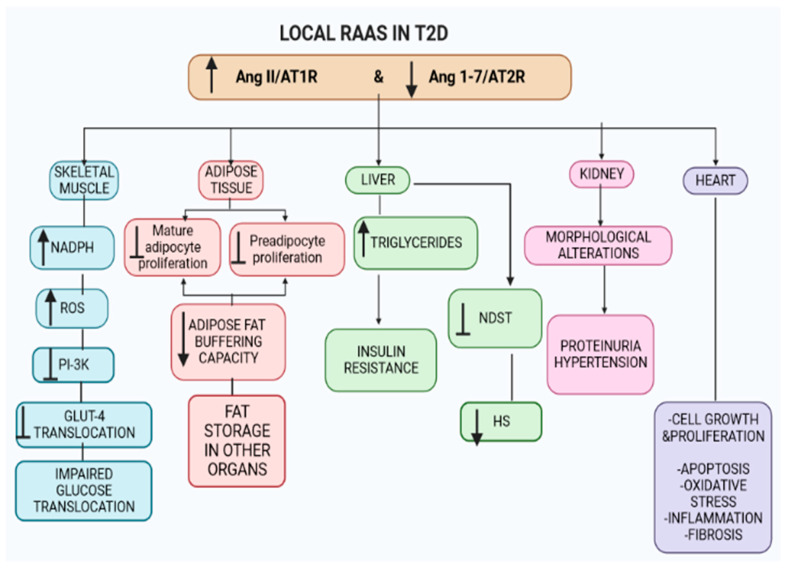
Local RAAS in various organs. Angiotensin II (Ang II) and angiotensin II type 1 receptor (AT1R) signaling are upregulated, whilst angiotensin 1-7 (Ang 1-7) and angiotensin type 2 receptor (AT2R) axis are downregulated in the numerous organs, i.e., skeletal muscle (leading to insulin resistance) and adipose tissue (decreasing the adipose tissue buffering capacity). The shift in the axis in the liver results in increased hepatic triglycerides, insulin resistance and inhibition of N-deacetylase (NDST), reducing heparan sulphate (HS). Additionally, the shift in Ang II/AT1R and Ang 1-7/AT2R in the kidneys and heart causes morphological changes, leading to organ dysfunction. **Key For all figures**. ↑ = increase/upregulation; ↓ = decrease; ┴ = inhibition; → = convert.

**Figure 2 ijms-24-11963-f002:**
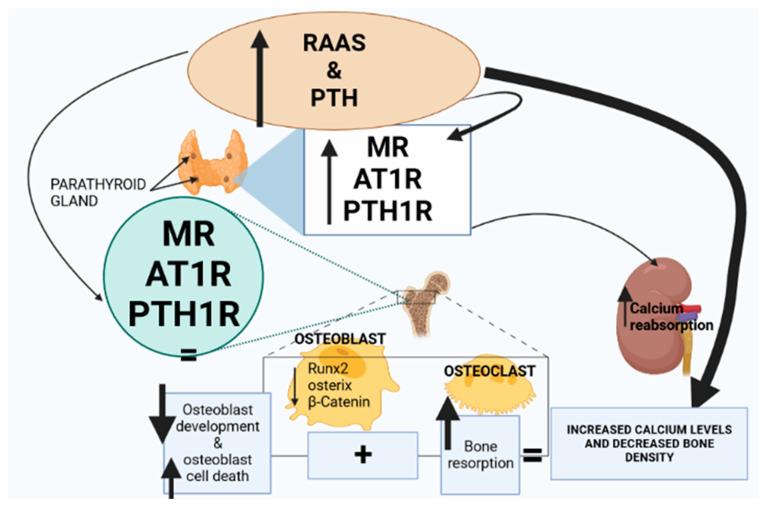
Renin–angiotensin–aldosterone system (RAAS) activity and parathyroid hormone (PTH) expression are elevated in T2D. The expression of mineralocorticoid receptors (MRs), angiotensin type 1 receptor (AT1R) and parathyroid hormone 1 receptor (PTH1R) is upregulated in the parathyroid gland and the bone marrow. Consequently, Runx, osterix and β-catenin are downregulated, leading to a decrease in osteoblasts; however, osteoclast activity is elevated. Furthermore, the upregulation of MRs, AT1R and PTH1R in the parathyroid glands stimulates PTH hypersecretion, leading to increased calcium reabsorption in the kidneys and collectively resulting in increased plasma calcium and decreased bone density.

**Figure 3 ijms-24-11963-f003:**
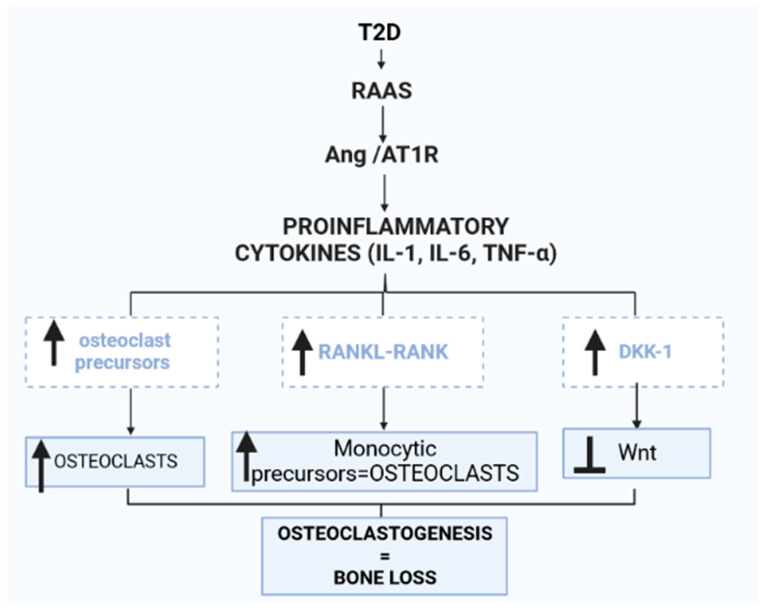
Renin–angiotensin–aldosterone system (RAAS) activity in the bone marrow is increased in type 2 diabetes (T2D). The Ang II/AT1R axis is upregulated, stimulating proinflammatory cytokines such as interleukin-1β (IL-1β), interleukin-6 (IL-6) and tumor necrosis factor-α (TNF-α). The increased activity of proinflammatory cytokines in the bone marrow increases osteoclast precursors, thus increasing osteoclasts. Additionally, the nuclear factor-κB ligand (RANKL) and nuclear factor-κB (RANK) interaction is enhanced, thus promoting the conversion of monocytic precursors to osteoclasts. Osteoclast differentiation is stimulated by M-CSF and RANKL. M-CSF induces the proliferation and survival of osteoclast precursor cells through activation of ERK and Akt. RANKL recruits TRAF6 to activate MAPKs, Akt and NFATc1 to promote the differentiation of osteoclast precursors to osteoclasts. Furthermore, increased proinflammatory cytokine activity in the bone induces the production of dickkopf-1 (Dkk-1), a wingless-related integration site (Wnt) antagonist that hinders the development of bone cells.

**Figure 4 ijms-24-11963-f004:**
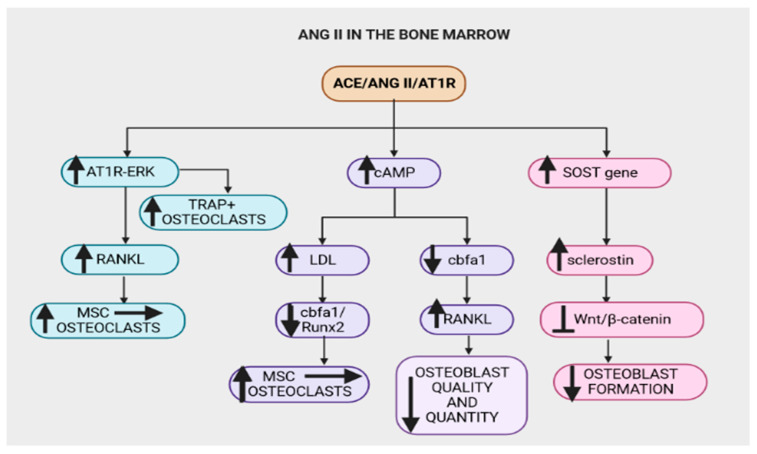
The Angiotensin-converting-enzyme (ACE)/angiotensin II (Ang II)/angiotensin type 1 receptor (AT1R) axis (ACE/Ang II/AT1R) in the bone marrow contributes to bone degradation by altering the microarchitecture of the bone structure. The ACE/Ang II/AT1R axis upregulates the AT1R–extracellular signal-regulated kinases (AT1R–ERK) pathway, which increases the formation of tartrate-resistant acid phosphatase positive (TRAP+) osteoclasts and nuclear factor-κB ligand (RANKL). RANKL promotes the conversion of mesenchymal stem cells (MSCs) to osteoclasts. Furthermore, this axis increases cAMP, downregulates core-binding factor subunit alpha-1/Runt-related transcription factor 2 (cbfα1/Runx2) and increases RANKL, subsequently decreasing osteoblast quantity and quality. Runx2 enhances the proliferation of suture mesenchymal cells and induces their commitment to osteoblast lineage cells. A decrease in Runx2 reduces the conversion of MSCs to osteoblasts, thus monocytic precursors are converted to osteoclasts. The ACE/Ang II/AT1R axis increases the expression of sclerostin coding gene (SOST) and hence the sclerostin protein that inhibits the wingless-related integration site (Wnt)/β-catenin pathway, thus significantly reducing osteoblast formation.

**Figure 5 ijms-24-11963-f005:**
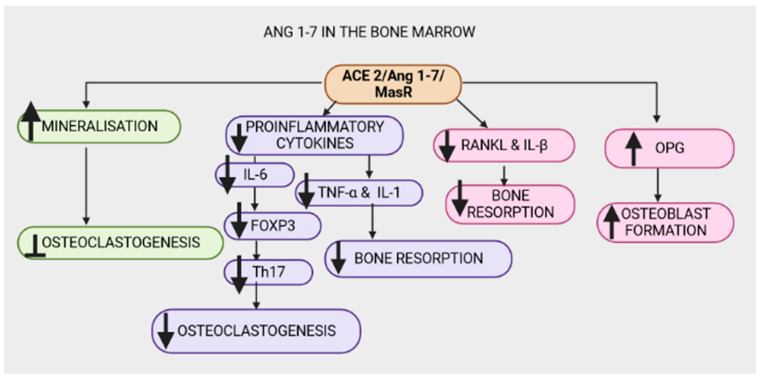
The Angiotensin-converting-enzyme 2 (ACE2)/angiotensin 1-7(Ang 1-7)/MasR (ACE2/Ang 1-7/MasR) axis exhibits osteoprotective properties in the bone marrow by increasing mineralization and inhibiting osteoclastogenesis. The ACE2/Ang 1-7/MasR axis counteracts the ACE/Ang II/AT1R signal by decreasing proinflammatory cytokines (interleukin-1 (IL-1β), interleukin-6 (IL-6) and tumor necrosis factor-α (TNF-α)). A decrease in IL-6 reduces the conversion of forkhead box P3 (FoxP3) T cells to T helper 17 (Th17) cells, consequently reducing osteoclastogenesis. Additionally, a reduction in the proinflammatory cytokines TNF-α, interleukin-1β (IL-1β) and nuclear factor-κB ligand (RANKL) decreases bone resorption. The ACE2/Ang1-7/MasR axis increases osteoprotegerin (OPG), resulting in increased osteoblast formation.

**Figure 6 ijms-24-11963-f006:**
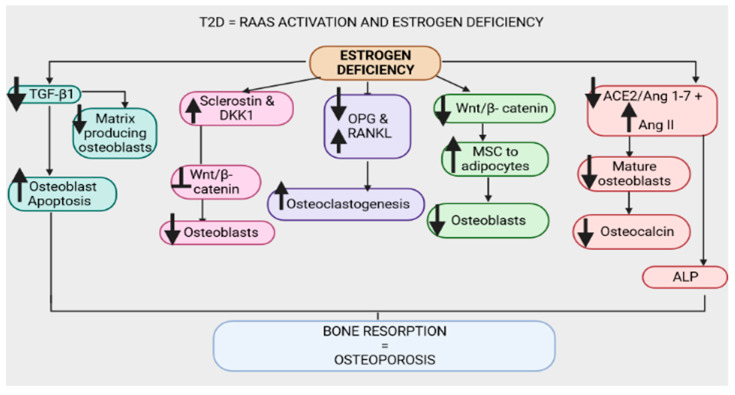
Upregulation of the renin–angiotensin–aldosterone system and estrogen deficiency have been evidenced in type 2 diabetes (T2D). Estrogen deficiency downregulates transforming growth factor β1 (TGF-β1), leading to enhanced osteoblast apoptosis and reduced matrix-producing osteoblasts. Furthermore, estrogen deficiency is associated with increased sclerostin and dickkopf-1 (Dkk-1), which inhibit the wingless-related integration site (Wnt)/β-catenin pathway, leading to a reduction in osteoblasts. Estrogen deficiency promotes the upregulation of nuclear factor-κB ligand (RANKL) and the downregulation of osteoprotegerin (OPG), leading to osteoclastogenesis and bone resorption. The wingless-related integration site (Wnt)/β-catenin pathway is downregulated in estrogen deficiency, thus enhancing the conversion of mesenchymal stem cells (MSCs) to adipocytes and reducing osteoblasts. Estrogen deficiency downregulates the angiotensin-converting-enzyme 2 (ACE2)/angiotensin 1-7 (Ang 1-7)/MasR (ACE2/Ang 1-7/MasR) axis and upregulates the angiotensin-converting-enzyme (ACE)/angiotensin II (Ang II)/angiotensin type 1 receptor (AT1R) axis (ACE/Ang II/AT1R), hence decreasing osteocalcin, alkaline phosphatase (ALP) activity and the number of mature osteoblasts.

## Data Availability

Not applicable.

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
