# Peer review of "The Relationship between Renin–Angiotensin–Aldosterone System (RAAS) Activity, Osteoporosis and Estrogen Deficiency in Type 2 Diabetes"

_ijms, 2023, doi:10.3390/ijms241511963_

Round 1

Reviewer 1 Report

In this review, Mkhize et al. provide an overview of the relationship between type 2 diabetes, osteoporosis, and the renin-angiotensin-aldosterone system (RAAS). Overall, the concept is good; however, there are several issues that need to be revised.

Introduction: This section provides a comprehensive overview of the renin-angiotensin-aldosterone system (RAAS) and its association with glucose homeostasis, estrogen deficiency, and osteoporosis. However, it could benefit from some improvements for clarity and readability.

·         Consider rephrasing some sentences to improve sentence structure and flow.

·         Break long sentences into shorter ones to enhance readability.

·         Provide more context for the significance and relevance of the relationship between RAAS and the mentioned conditions.

·         Consider reorganizing the content to present the information in a more logical order.

Section 'Ang I and Ang II via RAAS alternate pathways': The section discusses the link between RAAS upregulation, hyperglycemia, and insulin resistance. However, the presentation could be improved for clarity and coherence.

·         Simplify and clarify the explanations of the mechanisms and processes involved in RAAS activation.

·         Consider using subheadings to organize the content and make it easier to follow.

Section 'Local RAAS and T2D': The section provides information about the role of RAAS in the development and progression of insulin resistance in various organs. However, it could be improved.

·         Simplify complex explanations and use more concise language where possible.

·         Ensure clear transitions between ideas and concepts.

·         Consider using bullet points or numbered lists to present key points or factors.

Sections 'PRO-RESORPTIVE FACTORS': Overall comments:

·         Cite specific references: Instead of using general statements like "studies have shown" or "recent studies demonstrated," provide specific references or citations to support the claims made in the text. This adds credibility to the information presented.

·         Proofread for errors: Check for any grammatical errors, typos, or inconsistencies in punctuation and formatting. Ensuring proper proofreading will enhance the overall quality of the text.

Parathyroid hormone and Vitamin D3:

·         Use consistent and clear terminology: Maintain consistent terminology throughout the text. For example, the use of "calcium homeostasis" and "bone integrity" should be consistent and not interchanged.

·         Provide more context: In some parts, it would be helpful to provide additional context or background information to aid understanding. For instance, explain the significance of T2D (Type 2 Diabetes) in relation to calcium homeostasis and bone health.

Section 'Proinflammatory Cytokines':

·         Clarify the introduction: Begin the section with a brief introduction or sentence that sets the context for discussing the role of proinflammatory cytokines in bone homeostasis and disorders. This will provide better continuity with the previous sections.

·         Provide more explanation: Expand on the role of proinflammatory cytokines in bone remodeling. Explain how these cytokines attract osteoclast precursors and contribute to osteoclastogenesis, leading to bone loss. Additionally, describe the impact of proinflammatory cytokines on the Wnt signaling pathway and its effect on osteoblast function.

·         Review the figure description: The description of Figure 3 should provide a clear overview of the diagram and its relevance to the topic. Make sure it aligns with the content in the surrounding text and accurately describes the depicted processes and interactions.

Sections 'ACE/Ang II/AT1R axis and ACE2/Ang 1-7/Mas receptor and ANG 1-7':

·         Improve organization: Consider reorganizing the content to provide a clearer flow of information. Group related information together and use subheadings to distinguish different aspects of the ACE/Ang II/AT1R axis and the ACE2/Ang 1-7/Mas receptor axis.

·         Clarify the introduction: Begin the section with a brief introduction that explains the significance of discussing the ACE/Ang II/AT1R axis and the ACE2/Ang 1-7/Mas receptor axis in bone metabolism. This will help readers understand the context and relevance of the information.

Sections 'ANTI-RESORPTIVE FACTORS': Overall comments:

·         Improve clarity and sentence structure: Some sentences are long and complex, making it difficult to understand the information being conveyed. Simplify the language and sentence structure to improve clarity and readability.

·         Provide specific references: When mentioning studies or research findings, include specific references or citations to support the statements made in the text. This will enhance the credibility of the information and allow readers to explore the topic further.

·         Enhance organization: Consider reorganizing the content to provide a clearer flow of information. Group related information together and use subheadings to distinguish different aspects of the discussion.

·         Proofread for errors: Check for any grammatical errors, typos, or inconsistencies in punctuation and formatting. Ensuring proper proofreading will enhance the overall quality of the text.

Section 'Conclusion':

·         Clarify the objectives: In the conclusion, it would be helpful to explicitly state the main objectives or key findings of the review. This will provide a concise summary of the review's focus and contribution.

·         Expand on the study proposal: The last sentence of the conclusion mentions a study proposal, but it lacks specific details. Consider expanding on the potential study, including the research design, objectives, and expected outcomes.

Consider rephrasing some sentences to improve sentence structure and flow.

Break long sentences into shorter ones to enhance readability.

Simplify complex explanations and use more concise language where possible.

Proofread for errors: Check for any grammatical errors, typos, or inconsistencies in punctuation and formatting. Ensuring proper proofreading will enhance the overall quality of the text.

Reviewer 2 Report

Authors comprehensively reviewed the subject. With more than 200 references they did a hard work and gave a good overview. The plan is quite simple but efficient. The introduction is well written and in general there are a lot of details.However, although all sentences look good, the manuscript suffers flaws and shortcuts that tempered my enthusiasm.

Minor points: Some typos and grammar issues.

Title: do not use abbreviations in title “RAAS”

Line 93 : RAS or RASS?

Line 99: “the mRNA and protein expression of RAAS” what that means?

Line 222: a piece of the sentence is missing

Line 228: sentence is confusing, OB are not resorbing cells

Line 235: “attracting” may not be correct

Line 277: “a decline in bone density, increased risk of bone fracture and osteoporosis” / osteoporosis (cause) needs to appear before an increased risk of fracture (consequence)

Line 310:  “Therefore, Ang II modifies the Cbfa1 expression via the cAMP signalling pathway therefore reducing the number of osteoblasts, resulting in poor bone formation via stimulating the cAMP signalling pathway and regulating other down-stream targets such as LDL (111, 112).” Sentence structure needs improvement: twice cAMP signalling pathway…

Lines 334 to 337: already aforementioned…

Line 435 and before : IL1: which one?

Line 542: something is missing, the sentence has no meaning to me

Major points:

From line 117 to 190 : the impact of VD3 on osteoporosis needs to be reviewed carefully to avoid biased sentences.

Line 206: not consistent with above paragraph.

Line 244 – 246: RANKL does not drives pro-inflammatory cytokine expression. That’s the opposite.

Line 308: the link between LDL and CBFA1 needs improvement

Figures suffers concerns. I don’t know if it’s clumsy or wrong

-          Figure 1 :

o   needs to be completed with an opening on bone and bone cells

o   the resolution is too poor

o   check for arrows size and meaning

-          Figure 2:

o   Pieces are missing in the caption

o   Arrows remain inefficient to deliver a take home message

-          Figure 3:

o   Sounds like there is a confusion in the differentiation process of the osteoclast lineage, please redraw

-          Figure 4:

o   I guess it’s RANKL but not RANK

o   Check for MSC and hematopoetic lineage. Same as for fig3, looks like there is the misunderstanding on the role of Runx2 and other TFs in the differentiation of both OBs and OCs.

o   Check for arrows size

-          Figure 5:

o   IL-1 / Ilβ ???? please revised carefully

o   Please detail the causal relationship between mineralization and osteoclastogenesis: not clear

o   Why is osteoclastogenesis separated from bone resorption?

o   I know the link between OPG and OB but please detail the causal relationship between increased OPG and osteoblastogenesis: literature data on this?

-          Figure 6:

o   Osteocalcin comes after mature OBs

o   The result is an Unbalanced Bone remodeling in favor of bone resorption that leads to osteoporosis. Please correct.

Line 222: a piece of the sentence is missing

Line 542: something is missing, the sentence has no meaning to me

Reviewer 3 Report

Overall, this review paper provides a comprehensive overview of the relationship between the renin-angiotensin-aldosterone system (RAAS), osteoporosis, and type 2 diabetes (T2D). However, there are a few critical comments that could be considered in the revised manuscript:

1 Lack of context in the abstract: The abstract should provide a brief background and context for the study. It would be helpful to include a sentence or two explaining the significance of studying the RAAS, osteoporosis, and T2D relationship in terms of the broader field of research or clinical implications.

2 Limited discussion of mechanisms: While the abstract briefly mentions that the upregulation of RAAS may alter the bone microenvironment by shifting the OPG/RANKL ratio and inhibiting the Wnt/β-catenin pathway, it does not provide further elaboration or discuss other potential mechanisms involved in this relationship. Adding more details about these mechanisms and their relevance to T2D and osteoporosis would enhance the understanding of the topic.

3 Insufficient emphasis on human studies: The abstract mentions that RAAS upregulation has been positively correlated with T2D, estrogen deficiency, and postmenopausal osteoporosis. However, it does not specify whether these correlations were observed in human studies or in experimental models. Clarifying the evidence base and highlighting relevant human studies would strengthen the conclusions drawn in the review.

4 Lack of mention of potential therapeutic interventions: The abstract briefly mentions that research has focused on blocking RAAS to improve glucose handling, osteoporosis, and the downstream effects of estrogen deficiency. However, it does not discuss any specific therapeutic interventions targeting the RAAS pathway. Including a sentence or two about potential pharmacological or lifestyle interventions would provide a more comprehensive understanding of the practical implications of the findings.

5 Missing information on comorbidities: The abstract mentions that T2D is associated with a plethora of comorbidities, including osteoporosis. However, it does not specify which other comorbidities are commonly observed in T2D patients or their relevance to the RAAS-osteoporosis relationship. Providing a brief discussion of other important comorbidities associated with T2D and their potential interactions with RAAS and osteoporosis would enhance the scope of the review.

By addressing these points, the review paper could provide a more robust and informative overview of the relationship between RAAS, osteoporosis, and T2D.

fair

Round 2

Reviewer 1 Report

The authors have successfully addressed all the comments. I do not have any further concerns.

Reviewer 3 Report

The authors have addressed the comments that I raised in the first round review and I do not have further comments.